# A Comprehensive Approach to Bankruptcy Risk Evaluation in the Financial Industry

Samar Issa [1,*], Gulhan Bizel [2], Sharath Kumar Jagannathan [2] and Sri Sarat Chaitanya Gollapalli [2]

[1] Department of Business Administration, Saint Peter's University, Jersey City, NJ 07306, USA
[2] Data Science Institute, Saint Peter's University, Jersey City, NJ 07306, USA; gbizel@saintpeters.edu (G.B.); sjagannathan@saintpeters.edu (S.K.J.); sgollapalli@saintpeters.edu (S.S.C.G.)
[*] Correspondence: sissa@saintpeters.edu

**Abstract:** The study presents a comprehensive approach to examining the potential risk of bankruptcies in financial sector organizations. This investigation explores 20 financial sector entities and evaluates their fiscal history from 2000 to 2018. The developed model assesses the chance of these companies going bankrupt by analyzing indicators like liquidity, profitability, debt composition, and operational effectiveness. These metrics are contrasted to regulatory requirements and assessed as having low, moderate, or elevated risk repercussions, ultimately contributing to an overall threat rating. Additionally, the model has a unique algorithm that compensates for excessive debt levels, strengthening the reliability of the risk appraisal grade. This straightforward instrument illustrates the demand to incorporate a variety of financial health indicators. According to the findings, excessive amounts of debt have a detrimental influence on profitability, leading to decreased stock returns and a greater probability of bankruptcy. These findings have practical implications for investors and stakeholders, providing insightful information to help inform decision-making, especially during periods of economic unpredictability such as pandemics. Furthermore, they encourage the enhancement of financial market efficiency.

**Keywords:** financial ratios; risk assessment; bankruptcy; financial health; debt structure; profitability; liquidity; efficiency

## 1. Introduction

Global financial markets have undergone unprecedented volatility and have become highly uncertain in the contemporary era. The Global Financial Crisis (GFC) revealed that an ongoing increase in debt carries the potential for heightened financial instability or a complete debt crisis. This situation is further exacerbated by the complications resulting from the COVID-19 pandemic, which triggered economic contractions and financial challenges that have not yet been fully resolved.

Given the participation of majority and minority interests in a firm's trajectory, the capacity to foresee whether a corporation will face bankruptcy has grown in relevance. A company's failure impacts more than merely its shareholders; it has broader ramifications for the country's economy, involving supply chains, employment, and investor sentiment. Consequently, the ability to predict corporate insolvency has become a top priority for the academic and business sectors. Numerous research studies and models have evolved to determine if a company is in financial distress and on the verge of insolvency, capitalizing on technological developments in computing technology, data accessibility, and analytical resources. These initiatives encompass multiple models and approaches to detecting early warning indicators of financial problems within a corporation. This early detection enables proactive actions to address underlying concerns, thereby avoiding the path to bankruptcy.

Bankruptcy prediction in academic research employs a diverse range of methodologies, reflecting the complexity of financial dynamics. Traditional ratio analysis remains a foundational approach, scrutinizing key financial indicators such as liquidity, solvency, and profitability. Altman's Z-Score, a discriminant analysis model, is a classic example that assigns weights to various ratios to generate a composite score, aiding in classifying firms as financially distressed or healthy, and it is used in this paper. Logistic regression models extend this framework, accommodating a multitude of predictors, including financial ratios, market indicators, and macroeconomic variables. Artificial neural networks (ANN) and support vector machines (SVM) leverage advanced machine learning techniques to capture intricate non-linear relationships within financial data. Ensemble methods like Random Forest and Gradient Boosting amalgamate predictions from multiple models, enhancing predictive accuracy and mitigating the risk of overfitting. Additionally, time series analysis is employed to understand the temporal evolution of financial metrics, recognizing patterns and trends that may signify distress over time, and it is used in this paper to calculate excess debt. These diverse models collectively contribute to a robust and nuanced understanding of bankruptcy risk in academic research, offering insights into different facets of financial distress prediction.

This paper contributes to recent academic research on the topic of bankruptcy measures and the leveraging of banks in general. To undertake the empirical study on those issues, and in addition to the traditional ratio analysis, we introduced a new measure of overleveraging. If borrowing exceeds debt capacity, this can be called excess leveraging. Debt capacity is measured as sustainable or optimal debt. In other words, the measure of overleveraging is defined as the difference between actual and optimal debt. We followed the literature presented by Stein (2012) and Schleer and Semmler (2016) and focused on the solution of the dynamic version of the Stein model that allows us to use time-series data on banks in order to calculate the excess debt of twenty banks and financial firms.

Motivated by the above concerns, this paper estimates a model that assesses the financial firms' risk of insolvency based on their financial statements. Liquidity, solvency, efficiency, and profitability ratios are used to evaluate the company's financial health and its risk of bankruptcy. This paper adds to the literature by examining a new dataset comprised of 20 financial institutions for the time period 2000 to 2018. A comparative study of metrics with industry averages determined their respective performance. The company's ratios and its performance against industry benchmarks were used to create a risk score. Additionally, we have a unique indicator called "excess debt," which measures the financial performance of each company in the financial sector over the same time span and acts as a benchmark for the industry or as an early warning signal for a crisis. Adapting a post-GFC theoretical model of firm capital structure advanced by Stein (2012), this paper illustrates empirically that a firm's optimal debt is the debt capacity above which borrowing becomes risky. In this model, the optimal capital structure reflects the threshold beyond which firms' net worth declines. The estimation results surpass the traditional leverage calculation, i.e., the debt-to-equity ratio, a norm thus far, by adding major elements such as risk and return. As such, the contribution of this paper is manifold, engaging both with applied industry leveraging analysis common across financial markets.

The subsequent part of the paper is structured to offer an overview of both past and present accomplishments with a sophisticated machine-learning approach to bankruptcy prediction. Section 2 presents the literature review. Subsequently, the weighted average and risk score methodology are developed in Section 3. Section 4 presents empirical estimation results with some initial interpretations and the significance of bankruptcy prediction in financial decision-making along with macroeconomic instability and policy implications. Section 5 concludes the paper.

## 2. Literature Review

A visionary figure in the realm of financial analysis, Altman (1968) gained recognition for revolutionizing the way we assess companies' financial health. His role as an economist

responsible for assessing the company's performance involved liberating himself from the limitations of conventional financial metrics. Altman's journey began with a thorough examination of the analytical quality of ratio analysis, which demonstrated that it was an outmoded analytical approach. Altman used discriminant analysis to improve precision. In this instance, he used numerous economic parameters to create a more comprehensive model for predicting bankruptcy. The results from the ratio model were flawless for the first two years of operation. The calculation was invalid when the data increased for two years. Shabrina and Hadian (2021) dabbled in the discipline of dividend payout dynamics. The effects of current ratios, debt-to-equity ratios, and return on assets on dividend payments in mining businesses listed on the Indonesian stock exchange from 2016 to 2018 were researched. Despite recognizing the potential of more subtle indicators, their findings emphasized the significant impact of these characteristics). Irman and Purwati (2020) moved the emphasis to Indonesian Stock Exchange-listed automobile companies. The investigation focused on the impact of current ratios, debt-to-equity ratios, and total asset turnover on return on assets. While current ratios and total asset turnover had beneficial effects, debt-to-equity ratios had opposite effects. Separately, Purwanti and Warasto (2023) analyzed the financial data of Pempek Cawan Putih Restaurant from 2012 to 2021. Their investigation had quick ratios, current ratios, and cash ratios and yielded an intriguing result. While individual ratios had little influence on the profitability of assets, their aggregate impact was significant. The study's scope suggested an expansion into other industries. Yang and Kim (2020) investigated the elements that impact dividend payout ratios at state-owned banks. In their investigation, cash ratios, debt-to-equity ratios, total asset turnover ratios, return on assets, and business size took center stage. Surprisingly, only the company size was identified as a significant influence, casting doubt on the efficacy of other ratios. Another study by Yang and Kim (2020) investigated operating cash flow manipulation. Nariswari and Nugraha (2020) used multiple regression analysis to investigate the relationship between public debt servicing, exchange rates, and gross domestic product. Surprisingly, there was no evidence linking public debt servicing to economic growth. In another corner of financial research, the stock prices of Indonesian banking companies were under the microscope (Nariswari and Nugraha 2020). The study by Choiriyah et al. (2021) investigated the effect of operating profit margin and stock prices on the overall company's profitability and financial health. The stock prices of 32 banking firms listed on the Indonesian Stock Exchange were the focus of this study. Although only 8 out of the 32 companies satisfied the specified research hypothesis, when considered together, all firms had a significant influence on stock prices. The investigation used a multiple linear regression analysis, which revealed that the financial indicators Return on Assets (ROA), Return on Equity (ROE), Net Profit Margin (NPM), Earnings per Share (EPS), and Operating Profit Margin (OPM) had a significant effect on the company's profitability.

The capital structure of companies listed on the Indonesian Stock Exchange was observed to have less importance concerning Return on Assets and Total Asset Turnover. Nevertheless, when considered collectively, the Current Ratio, Return on Asset, Total Asset Turnover, and Sales Growth did affect the capital structure of these publicly listed companies. (Purba et al. 2020). In their quantitative study, Durrah et al. (2016) employed a purposive sampling method, gathering 12 samples from annual financial reports. The data collected underwent analysis through descriptive statistical techniques and multiple linear regression. The study's findings indicated that the Debt-to-Equity Ratio (DER) did not significantly impact stock prices. Conversely, Return on Equity (ROE) exhibited its influence on stock prices. Furthermore, both the DER and ROE influenced company stock prices.

The extant literature revealed financial analysis as a constantly evolving domain where conventional measures coalesce with inventive methods, constantly molding our comprehension of economic and corporate intricacies. In addition, non-standard amplification mechanisms such as the credit channel or financial stress on economic activity have recently started to become more important in theoretical modeling. The recent literature

concentrates on the banking sector as a source for business cycle dynamics. Such theoretical studies started with Stiglitz et al. (2003) and continued with Farmer and Geanakoplos (2009), Gorton (2010), Brunnermeier and Sannikov (2014), Mittnik and Semmler (2013), Issa (2020), Brunnermeier and Krishnamurthy (2020), and Issa and Gevorkyan (2022). The latter studies examine the balance sheets of banks and corporations, showing that a downward spiral is triggered through overleveraging, financial interdependencies, and contagion effects. This will motivate our empirical analysis, which focuses mostly on the debt side of the balance sheets.

The approach by Brunnermeier and Sannikov (2014) focuses specifically on the banking sector. In their view, it is a shock to asset prices, that creates a vicious cycle through the balance sheets of the banks. When there is low volatility in asset prices, the risk of excessive borrowing occurs. In other words, low volatility builds up instability. BS calls this the volatility paradox, which is when asset prices of assets held by banks fall, and the margin requirements for borrowing on the money market rise.

According to Mittnik and Semmler (2011), the vulnerability of the banks and the downward instability essentially depend on the improper incentive system and lack of constraints imposed on financial intermediaries, which allow for unconstrained growth of capital assets through borrowing. On the other hand, generous payouts affect banks' risk-taking, equity formation, and leveraging. Though, in the first instance, banks may have loan losses that might arise from defaults of firms or households, the foreign sector, or sovereign debt, the banks are substantially affected by financial stress, triggered by security price movements, high-risk premia, and credit spread, possibly exposed to a downward feedback loop.

According to Stein (2012), the destabilizing mechanism also results from a linkage between asset prices and borrowing. When assets are held by the banking system and tend to be overvalued, banks enjoy capital gains besides the normal returns, and they start to become overleveraged as compared to optimally leveraged. This happens when there are low financing costs for leveraging and there are capital gains emerging, providing the banks with high net worth. The actual operating income of banks is then composed of normal returns and stochastic capital gains. Debt tends to rise with capital gains and excess returns on capital, generating excessive borrowing.

In the model variant set out here, we follow both Brunnermeier and Sannikov (2014) and Stein (2012) overleveraging and excess debt approaches. In contrast to Mittnik and Semmler (2013), who work with nonlinear finance-macro links, we here, along the lines of both Brunnermeier and Sannikov and Stein, work with stochastic versions. In Brunnermeier and Sannikov as well as in Stein, the main force behind the instability of banking is overleveraging. The issue is, however, how overleveraging can also be measured and tracked empirically so that one obtains a clear early warning signal. Of course, the actual banking vulnerability will depend on other covariates than leveraging, such as a sudden rise in credit spreads, a rise in financial stress, and adverse feedback loops from economic activity to banks' balance sheets.

Finally, for the manuscript and detailed calculations, we follow Issa and Gevorkyan (2022) in calculating the optimal and excess debt. In their model, the optimal debt ratio maximizes the difference between net return and risk term. Therefore, only if the net return exceeds the risk premium does the optimal debt ratio become positive. The optimal debt ratio, therefore, is not a constant, as Stein (2012) noted, but rather varies directly with net return and risk. High risk implies high return; therefore, decreasing the bank's risk by providing secured lending to corporations will be a challenging task.

## 3. Methodology

Several previous studies used an empirical approach of initially choosing variables, followed by the Stepwise procedure to select the variables in the final analysis. However, these studies are limited in their ability to provide generalizable results as to what financial variables can consistently predict financial distress and bankruptcy risk. Therefore, in this

study, the following approach presented in Figure 1 is taken to select the best set of variables and risk measures for predicting bankruptcy. We start with a simple ratio analysis and a weighted average ratio to create an industry average score and a risk score. In addition, we used a novel metric called Excess Debt.

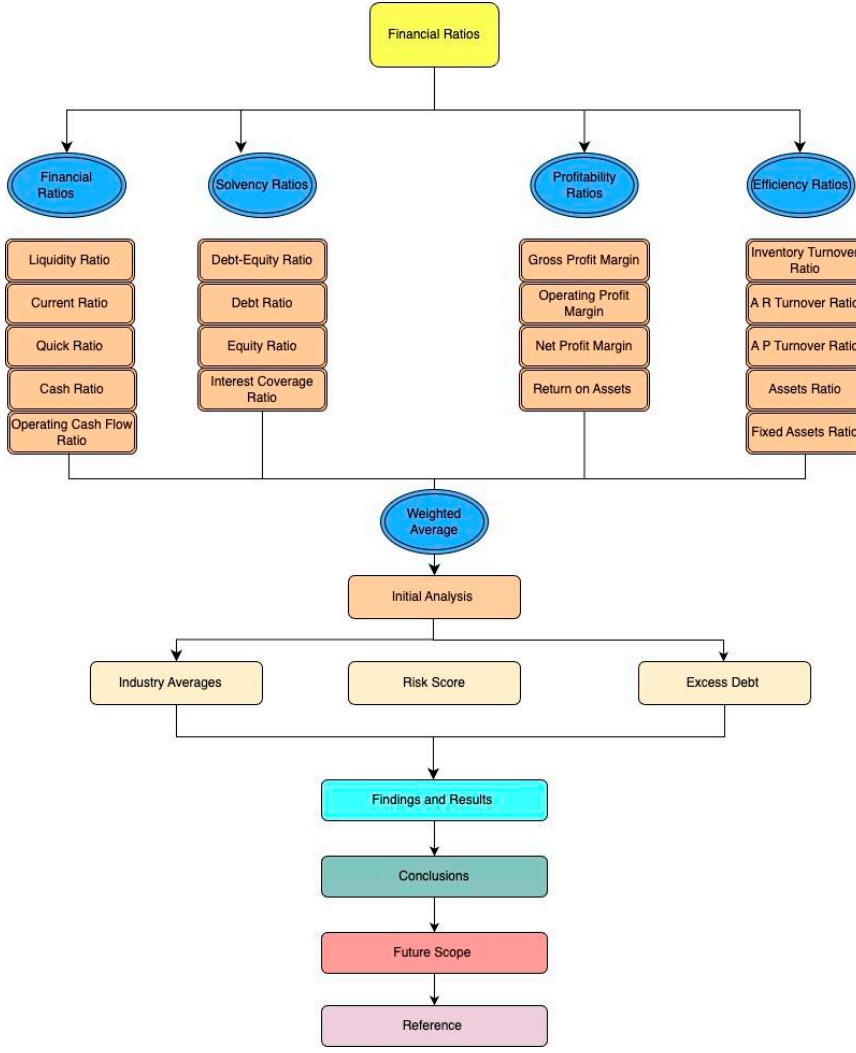

**Figure 1.** Flowchart of the methodology adopted for the research work.

As per Stein (2012), the GFC that engulfed the United States in 2008 was primarily a consequence of unsustainable debt-to-income ratios within private households, largely attributed to excessive financial obligations such as mortgages. Similarly, the financial crisis of the 1980s was closely tied to the business sector while also being intertwined with mounting levels of debt. Motivated by this, we calculated the excess debt and compared it to the industry average. The flow chart below illustrates the methodology adopted in this research paper.

### 3.1. Data Source

Our sample includes the top 20 publicly traded financial companies (depending on data availability) based on their market capitalization and total assets compared to others in the respective industry. The primary sources for corporate balance sheets data were Bloomberg Terminals and FactSet. We undertake our estimations using the full sample, which incorporates firm-specific variables on twenty publicly traded companies (depending on data availability) from the financial industry (as per the S&P500 ranking), based on their market capitalization and total assets compared to others in the respective industry.

The variables consist of balance sheets, income statements, and cash flow statements for the companies for the time period 2000 to 2018. The data was collected in April 2023. We capped it in 2018 to exclude COVID-19-era data, which consists of an external shock. Our focus is on the endogenous factors, which consist of an early warning sign of bankruptcy. The overall combined dataset has about 1100 rows, 25 columns, and 30,000 data values of interest. The statements contain key information needed to calculate the financial ratios that are the cornerstones of our analysis. Import fields include total revenue, operating profit, net income, total assets, total liabilities, and so on. The abbreviations of the companies for which data has been collected and analyzed are shown in Table 1:

**Table 1.** Sample companies and codes.

| Company Code | Company Name |
|---|---|
| BAC | Bank of America Corporation (Charlotte, United States) |
| BACHF | Bank of China Limited (Beijing, China) |
| BBAS3.SA | Banco do Brasil S.A. (Chiyoda City, Tokyo) |
| BCS | Barclays PLC (London, United Kingdom) |
| BNP.PA | BNP Paribas (Chiyoda City, Tokyo) |
| BRK-B | Berkshire Hathaway Inc. (Omaha, United States) |
| C | Citigroup Inc. ( New York, United States) |
| CS | Credit Suisse Group AG (Zürich, Switzerland) |
| DB | Deutsche Bank AG (Chiyoda City, Tokyo) |
| FMCC | Freddie Mac (Federal Home Loan Mortgage Corporation) (Virginia, United States) |
| FNMA | Fannie Mae (Federal National Mortgage Association) (Washington, D.C., United States) |
| GLE.PA | Société Générale S.A. (Paris, France) |
| GS | The Goldman Sachs Group, Inc. (New York, United States) |
| HSBC | HSBC Holdings PLC (London, United Kingdom) |
| JPM | JPMorgan Chase & Co. (New York, United States) |
| LYG | Lloyds Banking Group PLC (Edinburgh, United Kingdom) |
| NAB.AX | National Australia Bank Limited (Melbourne, Australia) |
| SMFG | Sumitomo Mitsui Financial Group, Inc.(Chiyoda City, Tokyo) |
| UBS | UBS Group AG (Zürich, Switzerland) |
| WFC | Wells Fargo & Company (New York, United States) |

*3.2. Financial Ratios*

Financial ratios are crucial in bankruptcy analysis as they offer a first-ground quantitative means to assess a company's financial health and potential distress. These ratios enable the early detection of financial warning signs, facilitate comparative analysis with industry peers, and reveal trends that may indicate deteriorating conditions. The financial ratios presented in Table 2 have been considered for our initial analysis as part of this research.

*3.3. Weighted Average*

A weighted average is a type of average that factors in not just the values being averaged but also the varying weights assigned to those values based on their relative importance or significance. This approach leans heavily on the values with higher weights in the calculation. The formula for computing a weighted average involves multiplying each value by its corresponding weight, summing these weighted values, and then dividing by the total sum of weights. This method is commonly used when some values hold more

weight or meaning in a particular context, such as in grading, financial analysis, statistics, or evaluating reviews (Ter Braak and Looman 1985). The weighted average can be calculated using the following formula, as defined in Equation (1).

$$Weighted\ Average = w1x1\ *\ w2x2\ *\ ....\ *\ wnxn \qquad (1)$$

**Table 2.** Financial Ratios calculated.

| Ratio | Category | Formula | References |
|---|---|---|---|
| Current Ratio | Liquidity Ratios (Makri et al. 2014) | $Current\ Ratio = \frac{Current\ Assets}{Current\ Liabilities}$ | Hantono (2018) |
| Cash Ratio | | $Cash\ Ratio = \frac{(Cash\ and\ Cash\ Equivalents)}{Current\ Liabilities}$ | Affandi et al. (2019) |
| Quick Ratio | | $Quick\ Ratio = \frac{(Current\ Assets - Inventory)}{Current\ Liabilities}$ | Purnomo (2018) |
| Debt to Equity Ratio | Solvency Ratios (Rahman 2017) | $D/E\ Ratio = \frac{Total\ Debt}{Shareholder's\ Equity}$ | Rajagukguk and Siagian (2021) |
| Debt Ratio | | $Debt\ Ratio = \frac{Total\ Debt}{Total\ Assets}$ | Kasasbeh (2021) |
| Debt to EBITDA | | $Debt\ to\ EBITDA\ Ratio = \frac{Total\ Debt}{Total\ EBITDA}$ | Anuar and Chin (2015) |
| Gross Profit Margin | Profitability Ratios (Rutkowska-Ziarko 2015) | $Gross\ Profit\ Margin = \frac{Gross\ Profit\ *\ 100}{Revenue}$ | Mahdi and Khaddafi (2020) |
| Net Profit Margin | | $Net\ Profit\ Margin = \frac{Net\ Income\ *\ 100}{Revenue}$ | Manglik and Goyal (2016) |
| Operating Profit Margin | | $Operating\ Profit\ Margin = \frac{Operating\ Profit\ *\ 100}{Revenue}$ | Handayani and Winarningsih (2020) |
| Return on Assets | Efficiency Ratios (Rashid 2017) | $Assets\ Turnover = \frac{Revenue}{Average\ Total\ Assets}$ | Vatansever and Hepsen (2013) |
| Receivables Turnover | | $Receivables\ Turnover = \frac{Net\ Credit\ Sales}{Average\ Accounts\ Receivable}$ | Hasanudin and Arviany (2022) |
| Payables Turnover | | $Accounts\ Payables\ Turnover = \frac{Total\ Purchases}{Average\ Accounts\ Payable}$ | Ilter (2020) |

The following weights have been considered for the individual ratios within the financial ratio categories in order to arrive at the categorical risk values as shown below in Table 3:

**Table 3.** Weights considered for individual financial ratios within each category.

| Ratio | Category | Weight |
|---|---|---|
| Current Ratio | Liquidity Ratios | 0.25 |
| Cash Ratio | | 0.5 |
| Quick Ratio | | 0.25 |
| Debt to Equity Ratio | Solvency Ratios | 0.34 |
| Debt to Capital Ratio | | 0.33 |
| Debt to EBITDA | | 0.33 |
| Gross Profit Margin | Profitability Ratios | 0.25 |
| Net Profit Margin | | 0.25 |
| Operating Profit Margin | | 0.5 |
| Return on Assets | Efficiency Ratios | 0.2 |
| Receivables Turnover | | 0.4 |
| Payables Turnover | | 0.4 |

The determination of weights for individual ratios within financial ratio categories is a crucial step in constructing a bankruptcy risk measure that emphasizes leverage risk. We have employed a combination of theoretical reasoning, empirical analysis, and expert judgment to assign weights to each ratio.

We provide here a brief review of books that can provide a theoretical foundation for the importance of these ratios.

The Principles of Managerial Finance by Gitman and Zutter (2012) discusses the role of the current ratio in assessing a company's short-term liquidity.

Financial Management: Theory & Practice by Brigham and Ehrhardt (2013) explains the cash ratio and its importance in evaluating a firm's ability to meet short-term obligations.

Fundamentals of Corporate Finance by Ross et al. (2013) offers insights into how the Debt-to-Equity ratio can be used to assess a company's financial leverage and long-term solvency.

Intermediate Accounting by Kieso et al. (2010) provides a comprehensive understanding of profitability ratios and their implications for assessing a firm's financial performance.

Financial Statement Analysis and Security Valuation by Penman (2010) explores the significance of return on assets in evaluating how efficiently a company uses its assets to generate earnings.

Fundamentals of Financial Management by Brigham and Houston (2012) discusses the importance of receivables and payables turnover in understanding a company's efficiency in managing its receivables and payables.

We also provide below a brief literature review of academic articles that support the choice of weights for financial ratios in a bankruptcy prediction model.

Alam et al. (2021) discuss various machine learning techniques for bankruptcy prediction, comparing different models and their accuracy. The paper provided insights into the effectiveness of specific financial ratios in predictive models.

Tian and Yu (2017) investigate the use of financial ratios for bankruptcy prediction across international markets, which offers a broader perspective on how ratios are weighted differently in various economic contexts.

Ohlson (1980) explores the ability of financial ratios to predict bankruptcy, employing a probabilistic approach. It is useful for understanding how different ratios can be weighted in terms of their predictive power, depending on the context.

In our model specifically, the weights reflect the theoretical significance of each ratio in capturing leverage-related risks. For example, within the Solvency Ratios category, the Debt to Equity Ratio, Debt to Capital Ratio, and Debt to EBITDA are assigned equal weights of 0.34, 0.33, and 0.33, respectively. This allocation stems from our view that these three ratios are equally important indicators of a company's leverage risk. These three ratios are equally important indicators of a company's leverage risk. This balance in weight distribution stems from the belief that no single ratio can fully capture the complexity of leverage risk, and a combination of these ratios provides a more comprehensive assessment.

Secondly, the weights are also influenced by empirical analysis and historical evidence. The assignment of a higher weight (0.5) to the Operating Profit Margin falls within the Profitability Ratios category since the literature values this ratio more heavily when assessing bankruptcy risk. This is based on observed correlations in historical data or empirical studies indicating that operating profit margin plays a significant role in predicting financial distress related to leverage.

Lastly, we have considered expert opinions and industry standards in determining the weights. The emphasis on the Cash Ratio with a weight of 0.5 within the Liquidity Ratios category places substantial importance on a company's ability to meet short-term obligations with cash. This emphasis was informed by expert opinions that stress the critical role of liquidity in mitigating leverage risk, especially during economic downturns or financial crises. The combination of these approaches contributes to a well-rounded and contextually relevant weighting scheme tailored to the specific focus on leverage-related bankruptcy risk.

### 3.4. Industry Averages

The industry average was used as the benchmark to compare the financial ratios of every company. For this purpose, we divided the data into four groups. The most recent years were given a higher weightage as this considers current performance while not ignoring past performance. Table 4 shows the weights for different groups:

**Table 4.** Weights for years as grouped above.

| Group | Year Range | Weight |
| --- | --- | --- |
| A | 2000–2006 | 0.1 |
| B | 2007–2010 | 0.25 |
| C | 2011–2014 | 0.25 |
| D | 2015–2018 | 0.4 |

The mean ratios for each of the categories were calculated, and then the weighted average was calculated for each ratio based on the categorical weights provided above. The resulting ratio value is the final ratio value for the company aggregated across the years 2000–2018. For the industry average, the average of the respective ratios for all the companies is taken, as shown in Table 5 below.

### 3.5. Risk Score

Based on a company's ratios and its performance against the industry averages, we calculated risk scores for each of the four categories of ratios: liquidity, solvency, profitability, and efficiency. Table 6 shows the categorical weights used for calculating the risk score. As seen, 60% of the weight is put on liquidity and solvency ratios. This is explained by the motivation of this paper, which emphasizes the role of leverage in debt-capital-intensive industries.

### 3.6. Excess Debt

We have considered the excess debt of each company over the period 2000–2018. Excess debt was defined by Stein (2012) as the difference between the actual and optimal debt. For the calculation of excess and optimal debt, please see Issa (2022). In each scenario, Stein made it clear that the primary source of the problem was not the presence of debt but excess debt within the country's firms under analysis. Stein's early warning signal of a debt crisis is based on the excess debt of households as a difference between the actual debt-to-assets ratio and the optimal debt ratio. Consequently, as the excess-debt ratio rises, the probability of a debt crisis increases. As the excess debt level rises, the probability of a debt crisis increases, as was the case in the 2008 crisis. The financial sector, by nature of the business and not necessarily by volatility indicators, has one of the highest excess debt ratios.

We calculate excess debt using the actual time series data of the twenty banks and financial institutions included in our sample. The first step is to compute the optimal debt. To derive an optimal-debt ratio, Stein (2012) used stochastic optimal control (SOC). A hypothetical investor selects an optimal-debt ratio, $f(t)$, to maximize the expectation of a concave function of net worth, $X(t)$, where T is the terminal date. The model assumes that the optimal debt-to-net-worth ratio significantly depends on the stochastic process concerning the capital-gain variable. The expected growth of net worth is also maximal when the debt ratio is at its optimal level.

**Table 5.** Company-wise average ratios based on a weighted average formula for the years 2000–2018.

| Company | Current Ratio | Cash Ratio | Quick Ratio | Debt to Equity | Debt to Capital | Debt to EBITDA | Gross Profit Margin | Net Profit Margin | Operating Profit Margin | Return on Assets | Receivables Turnover | Payables Turnover |
|---|---|---|---|---|---|---|---|---|---|---|---|---|
| BAC | 1.122543 | 0.015675 | 0.122543 | 8.34704 | 0.891012 | −434.415081 | 0.947001 | 0.686963 | 0.686963 | 0.947001 | 9.34704 | 1.333472 |
| BACHF | 1.034847 | 0.018743 | 0.077704 | 12.634245 | 0.885353 | 69.75049 | 1.236921 | 0.909727 | 0.971091 | 1.236921 | 13.644401 | 0.794255 |
| BBAS3.SA | 0.991008 | −0.01053 | 0.062437 | 14.340113 | 0.870108 | 63.729383 | 1.398779 | 0.969476 | 1.141491 | 1.398779 | 15.29208 | 0.336984 |
| BCS | 1.051727 | 0.011693 | 0.051727 | 24.058535 | 0.950938 | 678.557184 | 0.322253 | 0.131168 | 0.197106 | 0.322253 | 25.18758 | 0.584608 |
| BNP.PA | 0.989922 | 0.009946 | 0.047065 | 20.87265 | 0.89806 | 208.625464 | 0.474182 | 0.311222 | 0.337154 | 0.474182 | 21.882988 | 0.454119 |
| BRK-B | 1.867125 | 0.062436 | 0.867125 | 1.187171 | 0.537011 | 21.907178 | 4.450803 | 3.574291 | 3.651656 | 4.450803 | 2.202885 | 7.967669 |
| C | 1.115812 | 0.01516 | 0.115812 | 9.214086 | 0.896653 | 123.122939 | 0.927722 | 0.539549 | 0.531026 | 0.927722 | 10.219839 | 1.030738 |
| CS | 1.05187 | 0.009186 | 0.05187 | 22.054133 | 0.950727 | 100.863183 | 0.167087 | 0.046429 | 0.051684 | 0.167087 | 23.161672 | 0.100023 |
| DB | 1.038918 | 0.006084 | 0.038918 | 29.045025 | 0.962636 | 321.175025 | 0.082906 | 0.017856 | 0.020023 | 0.082906 | 30.057439 | 0.030328 |
| FNMA | 1.004853 | 0.002959 | 0.004853 | 136.267815 | 0.995391 | 102.723422 | −0.06601 | −0.168285 | −0.166861 | −0.06601 | 137.267786 | −0.005542 |
| GLE.PA | 0.972752 | 0.007179 | 0.044181 | 21.571433 | 0.886412 | 301.624122 | 0.354575 | 0.226163 | 0.247406 | 0.354575 | 22.576634 | 0.292807 |
| GS | 1.093948 | −0.00218 | 0.093948 | 11.538818 | 0.914435 | 86.991611 | 1.222843 | 0.818258 | 0.818258 | 1.222843 | 12.546141 | 1.259531 |
| HSBC | 1.078 | 0.010265 | 0.078 | 13.896378 | 0.92774 | 154.642294 | 0.715344 | 0.504883 | 0.546681 | 0.715344 | 14.950553 | 0.933851 |
| JPM | 1.103372 | 0.011858 | 0.103372 | 10.141996 | 0.906487 | 102.469021 | 1.22018 | 0.897707 | 0.893491 | 1.22018 | 11.166934 | 2.361287 |
| SMFG | 1.060791 | 0.027804 | 0.060791 | 21.580472 | 0.942739 | 275.728254 | 0.614943 | 0.344764 | 0.403439 | 0.614943 | 22.837469 | 2.260207 |
| LYG | 1.055112 | 0.002286 | 0.055112 | 19.739176 | 0.947887 | 53.452849 | 0.454095 | 0.301482 | 0.311294 | 0.454095 | 20.756619 | 0.555998 |
| NAB.AX | 1.064691 | 0.001444 | 0.064691 | 15.70598 | 0.939265 | 96.737306 | 1.018397 | 0.518596 | 0.717845 | 1.018397 | 16.714024 | 0.471372 |
| UBS | 1.050086 | 0.002261 | 0.050086 | 24.536475 | 0.952468 | 81.909643 | 0.342835 | 0.264542 | 0.270169 | 0.342835 | 25.60043 | 0.41997 |
| WFC | 1.114503 | 0.015841 | 0.114503 | 8.884382 | 0.897354 | 63.382606 | 1.738414 | 1.212675 | 1.231375 | 1.738414 | 9.889883 | 4.287679 |
| IA | 1.112068 | 0.011594 | 0.112068 | 22.656754 | 0.914568 | 131.679001 | 0.943936 | 0.647715 | 0.688134 | 0.943936 | 23.706456 | 1.371343 |

**Table 6.** Categorical weights used for calculating the overall risk score.

| Category | Weight |
|---|---|
| Liquidity Ratios | 0.4 |
| Solvency Ratios | 0.2 |
| Profitability Ratios | 0.3 |
| Efficiency Ratios | 0.1 |

Optimal leverage is given by:

$$f^*(t) = \left[ (r-i) + \beta - \alpha y(t) - \frac{\left(\frac{1}{2}\right)\left(\sigma_p^2 - \sigma_i \sigma_p \rho\right)}{\sigma^2} \right] \tag{2}$$

such that

$$Risk = \sigma^2 = \sigma_i + \sigma_p - \left(2\rho_{ip}\sigma_i\sigma_p\right)$$

where $r$ is the bank's capital gain/loss; $i$ is the credit cost of banks; $\beta$ is the productivity of capital; $y(t)$ is the deviation of capital gain from its trend; $\sigma^2$ is the variance; and $\rho$ represents the negative correlation coefficient between interest rate and capital gain. Through the presented model, Stein could determine excess debt and an early warning signal of a potential crisis. As mentioned, it is this mechanism that played a role in the decreasing net worth of individuals, households, and institutions in the United States, and that was amplified by the increased leverage and pricing volatility of complex securities.

To measure the excess leveraging of banks, the introduced and defined Stein model was followed with a focus on the solution of the dynamic version of the model, which allowed for the use of time-series data on banks. One difference from Stein is that, in this case, each bank's productivity of capital was not assumed to be deterministic or constant as in the Stein model; rather, it was calculated for the years 2000–2018.

The optimal debt level was calculated for the years 2000-2018; thus, excess debt, which is the measure of overleveraging in this paper, was estimated. To calculate the firm's optimal debt ratios, data on the firm's capital gain/loss, market interest rates, and the productivity of capital were collected. Using these variables, the risk and return components of the model were then calculated. Using the abovementioned variables, the optimal and actual debt ratios were calculated for a sample.

Equation (2) provide us with the dynamics of optimal debt. By applying those dynamics to actual time series data, Stein (2012) provides an explicit solution for the empirical computation of optimal debt and excess debt.

In our computation of capital gains, we use the HP-filter and employ the trend from the HP-filter as capital gains. Capital gains are measured using the banks' market equity valuation. The return, beta, is obtained through the gross income of the bank over capital (capital = equity long-term debt+ short-term debt/2), and the interest payments are obtained from the income statement of the banks. Volatility is measured as the moving average over 3 quarters. The actual debt over net worth is computed by the actual debt divided by total assets. Then both the optimal and actual debts are normalized. For more details on excess debt across industries, please see Issa and Gevorkyan (2022).

Following the methodology mentioned above, we estimate the optimal leverage for our industry sample. The analysis was performed using the total long-term debts and total assets. As noted, total long-term debt represents the company's total debt with a maturity date more than one year from the balance sheet date.

The vertical axes of Figures 2 and 3 represent the debt ratios for each company, while the horizontal axes represent the years. Figure 2 shows the optimal debt against the actual debt ratios for the financial industry, and Figure 3 shows the excess debt for the same time period. Please note that the excess debt is the difference between optimal and actual debt.

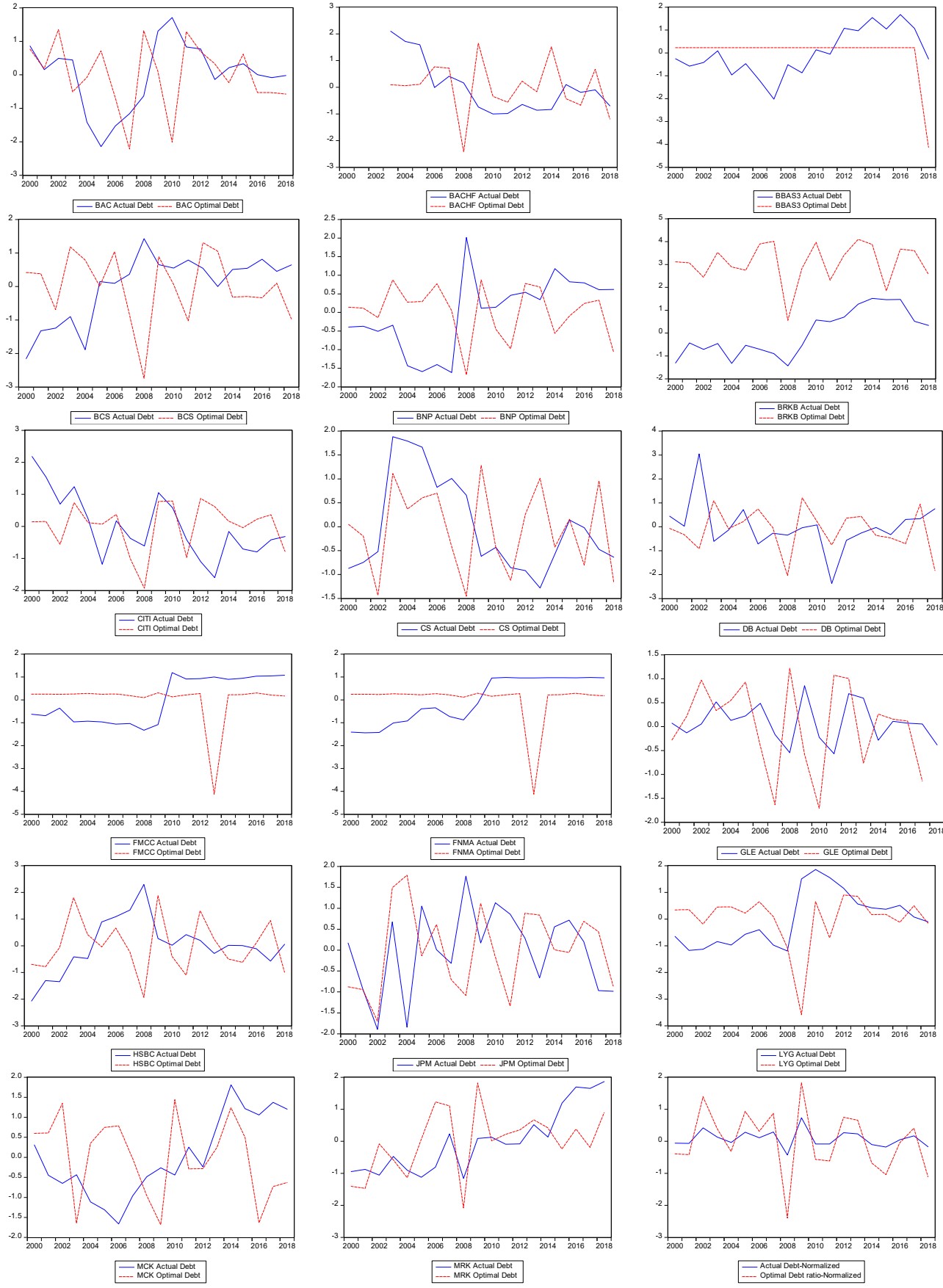

**Figure 2.** *Cont.*

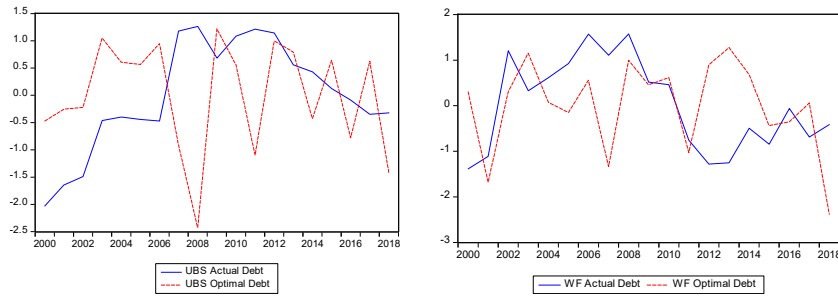

**Figure 2.** Actual vs. optimal debt by company estimation results. Source: authors' calculations.

**Figure 3.** *Cont.*

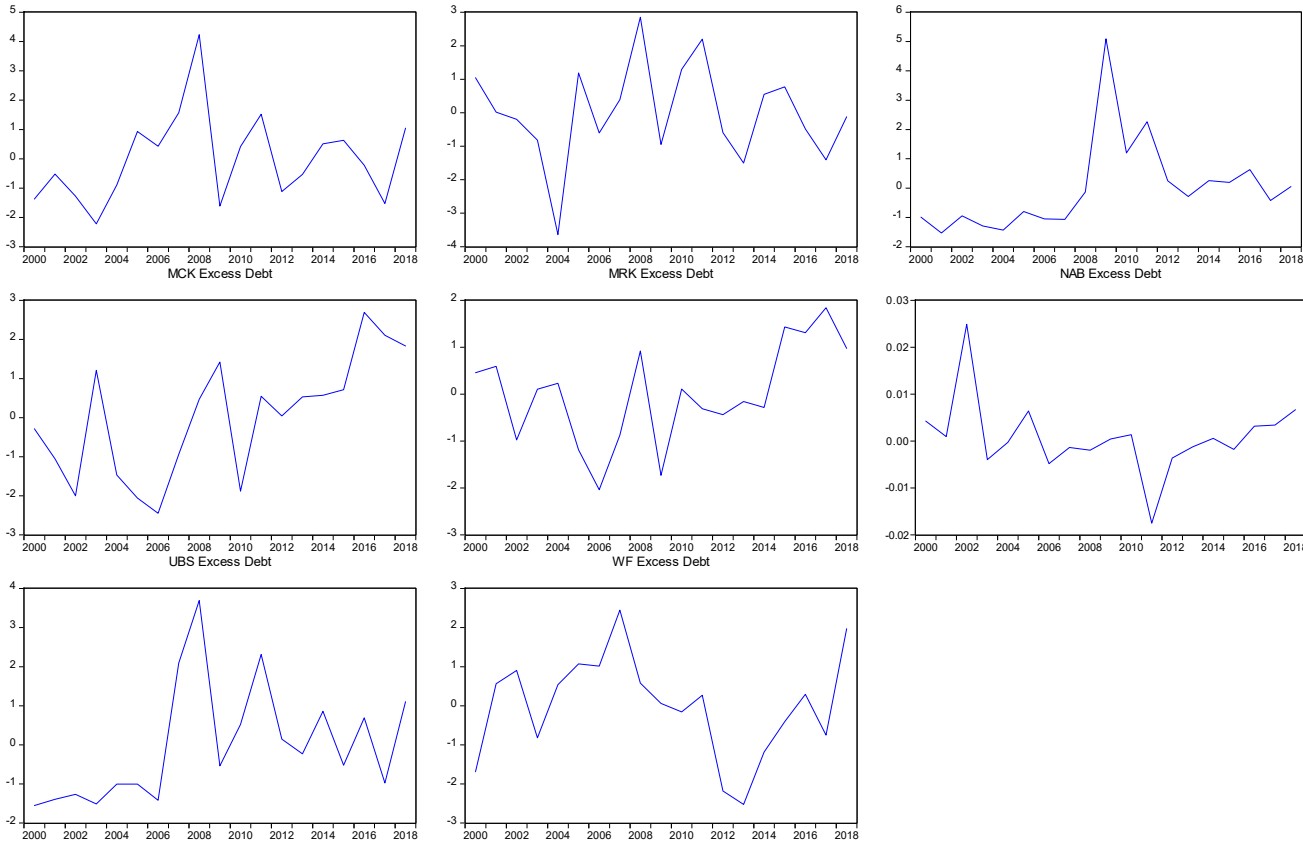

**Figure 3.** Excess debt by company estimation results. Source: authors' calculations.

The optimal and actual debt ratios for most of the firms exhibited similar trends. For several years preceding the 2007–2009 GFC, corporations had high optimal debt ratios. For most of the firms, about one or two years prior to 2007, the optimal debt ratios began to drop, and the decline was severe in all cases. The trend of actual debt exceeding the optimal debt reversed post-GFC for most industries in the sample.

We also calculate each company's actual debt ratio. Prior to the GFC, the data points to a rising actual debt ratio in the years prior to the GFC.

Analyzing the excess debt and leverage of each bank involves examining its financial health, risk factors, and broader economic implications. We briefly discuss each bank and financial institution in our sample below.

First, Bank of America faced challenges during the 2008 GFC, primarily due to its acquisition of Merrill Lynch and Countrywide Financial, leading to substantial losses. The excess debt during this period contributed to a need for government assistance through the Troubled Asset Relief Program (TARP). In addition, Citigroup was significantly affected by the 2008 GFC, requiring substantial government support. Excessive leverage and exposure to risky assets contributed to its difficulties (Bank of America 2022a, 2022b).

Goldman Sachs faced criticism for its role in the subprime mortgage market. High leverage ratios and a shift towards riskier assets impacted the firm. Similarly, Wells Fargo faced scrutiny for its sales practices scandal, impacting its reputation. Excessive risk-taking and aggressive sales goals contributed to the bank's challenges.

JPMorgan, while relatively resilient during the GFC, faced challenges related to the "London Whale" trading incident. The bank's diversified business model helped mitigate some risks associated with excessive leverage.

UBS also faced challenges during the GFC, particularly due to its exposure to subprime mortgages. The bank underwent significant restructuring to address its risk profile.

For Freddie Mac and Fannie Mae, both entities played a significant role in the GFC due to their exposure to subprime mortgages. The government placed them into conservatorship to prevent a collapse that could have had severe economic consequences.

On the other hand, Berkshire Hathaway, led by Warren Buffett, generally maintains a conservative financial approach. The company's leverage and debt levels are often lower compared to traditional financial institutions.

Chinese banks, including the Bank of China, have historically operated with high levels of debt to support rapid economic growth. The Chinese government has taken measures to address concerns about the banking sector's excessive debt and mitigate systemic risks.

European banks faced challenges during the Eurozone debt crisis. For instance, Barclays faced scrutiny for its actions during the GFC, including raising capital from Middle Eastern investors. High leverage ratios contributed to regulatory concerns and changes in leadership.

Lloyds faced challenges during the GFC due to its acquisition of HBOS. Government intervention and capital injections were necessary to stabilize the bank.

BNP Paribas also faced challenges and was excessively leveraged, but mostly the exposure to sovereign debt was what impacted its financial stability.

Credit Suisse faced challenges related to its exposure to mortgage-backed securities and legal issues. High leverage ratios and risk concentrations impacted its financial performance.

Deutsche Bank faced various challenges, including legal issues, regulatory fines, and concerns about its financial health. Excessive leverage and exposure to risky assets were contributing factors.

HSBC has a global presence and has faced challenges related to its exposure to subprime mortgages and money laundering issues. The bank's extensive global operations contributed to its complexities.

Société Générale faced challenges related to the GFC and a trading scandal. Excessive leverage and inadequate risk management were contributing factors.

Brazilian banks, including Banco do Brasil, have faced challenges related to economic volatility and political instability. The impact of excessive debt is influenced by the country's economic conditions.

Australian banks generally weathered the 2008 GFC well, with National Australia Bank being no exception. The impact of excessive debt was moderated by Australia's resilient economy.

Japanese banks, including Sumitomo Mitsui Financial Group, have historically had conservative lending practices. While the Japanese banking sector faced challenges, excessive debt was less pronounced compared to some Western counterparts.

Each of these cases illustrates how different financial institutions were impacted by and responded to the challenges of the financial crisis, with varying degrees of government intervention and restructuring efforts. The experiences of these institutions highlight the importance of risk management and the potential consequences of excessive leverage and exposure to high-risk assets.

To establish a relationship between the risk score and excess debt, we have considered the number of years for which the company's excess debt outperformed the industry as a whole. Depending on how many years the company outperformed or underperformed against the industry average, these values were used to categorize the risk from excess debt as Low, Medium, or High, as shown in Table 7.

Based on the results obtained, it may be argued that the financial industry is relatively vulnerable. The financial industry, in general, is seen as the driving force behind the 2008 GFC, but some had smooth behavior afterward. What we can see from individual banks is that the larger the bank, the higher the excess debt. We see GLE.PA, GS, BRK-B, SMFG, C, and NAB.AX as having a high-risk score or bankruptcy risk. The rest of the financial firms showed a medium risk score. Designing an effective policy to make optimal debt a fixed ratio based on the net worth of a financial corporation is a regulatory challenge.

**Table 7.** Categorical scores for the companies being analyzed.

| Company | lscore | sscore | pscore | escore | Overall Risk | <Industry Average | >Industry Average | ED Risk |
|---------|--------|--------|--------|--------|--------------|-------------------|-------------------|---------|
| BAC | 30 | 10 | 83.75 | 68 | 45.925 | 10 | 8 | Medium |
| DB | 90 | 66.95 | 114 | 42 | 87.79 | 10 | 8 | Medium |
| UBS | 90 | 10 | 114 | 48 | 77 | 11 | 7 | Medium |
| LYG | 90 | 10 | 114 | 58 | 78 | 12 | 6 | Medium |
| JPM | 75 | 10 | 30 | 42 | 45.2 | 9 | 9 | Medium |
| HSBC | 67.5 | 10 | 90 | 76 | 63.6 | 11 | 7 | Medium |
| BACHF | 47.5 | 10 | 30 | 76 | 37.6 | 9 | 9 | Medium |
| FNMA | 90 | 38.9 | 114 | 42 | 82.18 | 10 | 8 | Medium |
| CS | 80 | 10 | 114 | 58 | 74 | 10 | 8 | Medium |
| BNP.PA | 80 | 38.05 | 114 | 58 | 79.61 | 9 | 9 | Medium |
| BCS | 80 | 38.05 | 114 | 48 | 78.61 | 10 | 8 | Medium |
| BBAS3.SA | 90 | 10 | 12 | 76 | 49.2 | 10 | 8 | Medium |
| WFC | 37.5 | 10 | 12 | 42 | 24.8 | 11 | 7 | Medium |
| C | 50 | 10 | 105 | 76 | 61.1 | 8 | 10 | High |
| GLE.PA | 90 | 38.05 | 114 | 58 | 83.61 | 5 | 13 | High |
| GS | 85 | 10 | 30 | 68 | 51.8 | 7 | 11 | High |
| BRK-B | 10 | 10 | 12 | 42 | 13.8 | 8 | 10 | High |
| SMFG | 47.5 | 38.05 | 114 | 24 | 63.21 | 7 | 11 | High |
| NAB.AX | 90 | 10 | 66.25 | 68 | 64.675 | 7 | 11 | High |

The analysis provided above touches on several key aspects of financial risk and management, particularly in the context of the GFC. It aligns with a range of academic research that has explored similar themes. The discussion about the trends in optimal and actual debt ratios before and after the GFC resonates with the research focus of Edward I. Altman, who is renowned for his work on bankruptcy prediction models. The specific challenges faced by banks like Bank of America, Citigroup, and others, as well as their responses to these challenges, are reflective of broader research themes in financial economics. The issues of excessive leverage, risk-taking, and government interventions are central to studies on financial stability and crisis management. The focus on risk management practices, as well as the financial health of banks, aligns with current academic discourse that explores the relationship between risk management strategies, leverage, and financial stability. This includes research on how banks manage risks associated with lending practices, exposure to high-risk assets, and the overall stability of their balance sheets.

*3.7. Adjusted Risk Score and Bankruptcy Probability*

Adjusted Risk Score refers to the overall risk score that was calculated and adjusted by considering the levels of Excess Debt risk obtained above. The "Low", "Medium", and "High" categories were given certain values in order to convert them to a meaningful score level for Excess Debt risk. This was then used along with the overall risk score to arrive at an "Adjusted Risk Score" that incorporates the Excess Debt Risk with the Overall Risk Score. The mean score is obtained by taking the average of this adjusted risk score for all the companies. This mean score is used to determine the expected probability level for a company to go bankrupt, with the possible values being either low or high, as shown in Table 8.

**Table 8.** Bankruptcy prediction table for the companies analyzed.

| Company | Liquidity Risk Score | Solvency Risk Score | Profitability Risk Score | Efficiency Risk Score | Overall Risk Score | #Months Score <Industry Average | #Months Score >Industry Average | Excess Debt Risk | Adjusted Risk Score | Bankruptcy Probability |
|---------|---------------------|---------------------|--------------------------|-----------------------|--------------------|--------------------------------|--------------------------------|------------------|---------------------|------------------------|
| BAC | 37.5 | 10 | 60 | 85 | 43.5 | 10 | 8 | Medium | 43.5 | Low |
| BNP.PA | 80 | 38.05 | 95 | 85 | 76.61 | 9 | 9 | Medium | 76.61 | Low |

**Table 8.** *Cont.*

| Company | Liquidity Risk Score | Solvency Risk Score | Profitability Risk Score | Efficiency Risk Score | Overall Risk Score | #Months Score <Industry Average | #Months Score >Industry Average | Excess Debt Risk | Adjusted Risk Score | Bankruptcy Probability |
|---|---|---|---|---|---|---|---|---|---|---|
| SMFG | 47.5 | 38.05 | 95 | 42.5 | 59.36 | 7 | 11 | High | 74.2 | Low |
| BRK-B | 10 | 10 | 10 | 52.5 | 14.25 | 8 | 10 | High | 17.8125 | Low |
| GS | 85 | 10 | 35 | 85 | 55 | 7 | 11 | High | 68.75 | Low |
| C | 37.5 | 10 | 75 | 85 | 48 | 8 | 10 | High | 60 | Low |
| WFC | 37.5 | 10 | 10 | 52.5 | 25.25 | 11 | 7 | Medium | 25.25 | Low |
| BBAS3.SA | 90 | 10 | 19 | 95 | 53.2 | 10 | 8 | Medium | 53.2 | Low |
| BCS | 67.5 | 38.05 | 95 | 72.5 | 70.36 | 10 | 8 | Medium | 70.36 | Low |
| DB | 90 | 38.05 | 95 | 60 | 78.11 | 10 | 8 | Medium | 78.11 | Low |
| CS | 80 | 10 | 95 | 85 | 71 | 10 | 8 | Medium | 71 | Low |
| HSBC | 80 | 10 | 75 | 95 | 66 | 11 | 7 | Medium | 66 | Low |
| BACHF | 47.5 | 10 | 25 | 95 | 38 | 9 | 9 | Medium | 38 | Low |
| FNMA | 90 | 38.9 | 95 | 52.5 | 77.53 | 10 | 8 | Medium | 77.53 | Low |
| JPM | 62.5 | 10 | 25 | 52.5 | 39.75 | 9 | 9 | Medium | 39.75 | Low |
| LYG | 90 | 10 | 95 | 85 | 75 | 12 | 6 | Medium | 75 | Low |
| UBS | 90 | 10 | 95 | 72.5 | 73.75 | 11 | 7 | Medium | 73.75 | Low |
| GLE.PA | 90 | 38.05 | 95 | 85 | 80.61 | 5 | 13 | High | 100.7625 | High |
| NAB.AX | 90 | 10 | 56.25 | 95 | 64.375 | 7 | 11 | High | 80.46875 | High |

## 4. Findings and Results

The results from the above estimations suggest that the estimated excess debt for most companies in the financial sector in the US has largely been moving up, spiking around the GFC, and then declining, only to resume the rise again later. Despite that, most of the companies exhibited a low risk of bankruptcy, except for Société Générale and National Australia Bank Limited. Let us examine the case of GLE.PA (Société Générale S.A.—Bankruptcy Probability—High) to try and understand how the process works.

GLE.PA has a liquidity score of 90, a solvency score of 38, a profitability score of 95, and an efficiency score of 85. The company has had high liabilities since 2004, while it also saw a steep increase in its revenue until 2006. After the GFC in 2008, revenue started to decrease while the liabilities continued to rise. Looking at Figure 4, we can observe that this has been the case until 2018. Although the company has seen a recovery in revenues, the recovery has not reached the pre-GFC levels. Since we are giving higher priority to the current performance, the scores calculated will give a red flag for the increasing leverage and not-so-higher rate of increase in revenue. Until the company works on reducing its debt to a sustainable level while improving its profitability, the company will continue to be at a high risk of going bankrupt.

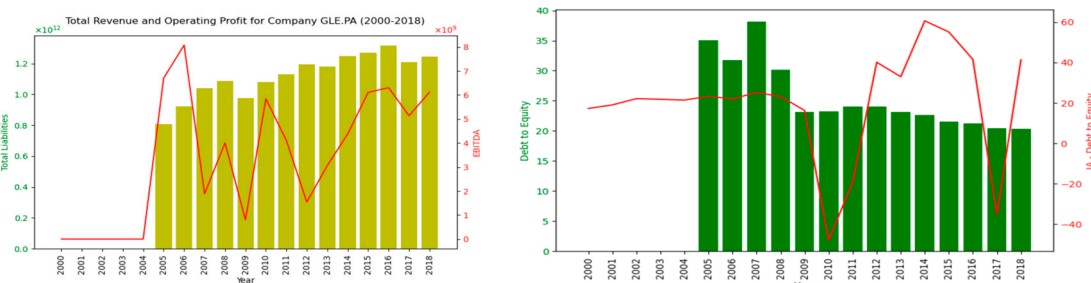

**Figure 4.** Total Liabilities vs. EBITDA and Debt-to-Equity Ratios for GLE.PA over the period 2000–2018.

More broadly, Figure 5 adds to our discussion about recent observations on the actual debt evolutions in the U.S. corporate sector. There are signs that overleveraging across industries is likely on the rise again (Vandevelde 2020). At first, it might be tempting to view the evidence in this paper as industry-specific and relevant to the isolated debt risk

assessments. For example, the financial sector is caught up in the unpredictability of risk valuations and future cash flows.

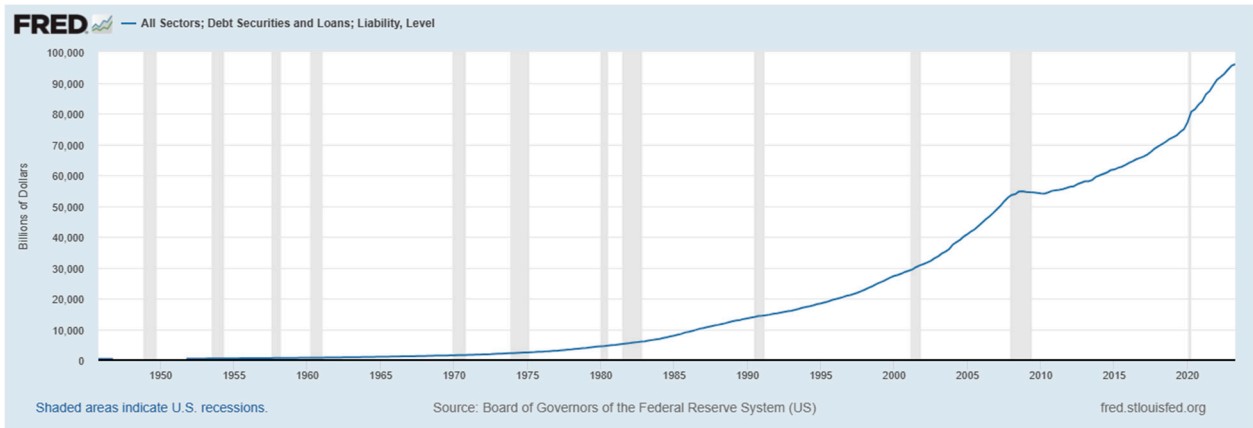

**Figure 5.** Actual debt evolution in the U.S. corporate sector.

From our estimation, we can see that many financial companies in the United States, such as banks and financial institutions, often have high levels of debt on their balance sheets, but they can still maintain a low risk of bankruptcy due to several key factors. For instance, financial companies are subject to extensive regulations and oversight by government agencies such as the Federal Reserve, the Office of the Comptroller of the Currency (OCC), and the Federal Deposit Insurance Corporation (FDIC). These regulations are designed to ensure the stability and soundness of the financial system. Regulatory agencies impose capital adequacy requirements, stress tests, and other measures to ensure that banks have sufficient capital to absorb losses and withstand economic downturns. In addition, many large financial institutions have diverse revenue streams and business lines, which can help mitigate the risks associated with any single financial product or service. Financial companies also often have access to a wide range of funding sources, including deposits, short-term and long-term debt, and equity capital. This diversity of funding sources allows them to adjust their capital structure as needed and reduce reliance on a single source of funding. It should be noted that financial companies typically have sophisticated risk management practices in place. They employ risk assessment models, stress testing, and hedging strategies to manage and mitigate financial risks, such as credit risk, market risk, and interest rate risk. Finally, economic conditions can also play a significant role. During periods of economic growth, financial institutions tend to perform well, generating profits that can be used to service their debt. Strong economic conditions contribute to low default risk.

## 5. Conclusions

The risk measure presented in this paper is an important metric that can help corporations detect the risk of bankruptcy. The paper relies on a complex model that considers key financial ratios and uses historical data to calculate these ratios and an associated risk score. The weighted average helps give preference to the current performance of the company of interest while at the same time giving some consideration to its performance during testing times like the 2008 GFC. Moreover, the optimal debt ratio estimation presented in this paper is an important measure that can help financial firms detect a sustainable debt level above which it becomes risky to leverage. This is a key strength of this model, as it helps understand a company's ability to make it through challenges unscathed and use this knowledge to predict bankruptcy with higher accuracy.

We have seen that financial companies rely heavily on debt but still maintain financial stability during steady economic periods. According to the findings, excessive amounts of debt have a detrimental influence on profitability, leading to decreased stock returns and a

greater probability of bankruptcy. Moreover, excessive debt and leverage in major financial institutions can contribute to systemic risk, potentially leading to a cascading effect on the global financial system.

These findings have practical implications for investors and stakeholders, providing insightful information to help inform decision-making, especially during periods of economic unpredictability such as pandemics. Furthermore, they encourage the enhancement of financial market efficiency.

The results of the paper lead to some policy implications helpful to improving the performance of the modern financial industry. It is important to note that the financial industry is complex and highly regulated, and the factors contributing to the low risk of bankruptcy for financial companies can vary by institution. While these factors can help mitigate the risk of bankruptcy, it is essential for regulators, institutions, and investors to remain vigilant and proactive in managing and accessing financial risks to maintain the stability of the financial system.

While the study provides a robust analysis of the potential risks of bankruptcies in the financial sector, several limitations merit consideration. Firstly, the study's sample size of 20 financial sector entities may be insufficient to capture the diversity and complexity of the entire industry. The findings may lack generalizability, and the specific characteristics of the chosen entities may not be representative of broader trends within the financial sector. A more extensive and varied sample would enhance the external validity of the study, allowing for a more accurate assessment of the factors influencing bankruptcy risk across a spectrum of financial organizations.

Secondly, the study's reliance on historical fiscal data from 2000 to 2018 introduces a temporal limitation. Economic conditions, regulatory landscapes, and market dynamics can evolve over time, potentially impacting the relevance of historical patterns in predicting future bankruptcies. The study would benefit from an exploration of the model's performance across different time periods or the inclusion of more recent post-COVID-19 data to assess the stability and applicability of the developed model in the face of changing financial environments. Addressing these limitations would strengthen the study's foundation and contribute to a more nuanced understanding of the dynamics surrounding bankruptcy risks in the financial sector.

The research has a wide scope for future work. We have numerous models and approaches to predicting bankruptcy. The current model focuses on giving higher preference to the latest years while at the same time giving weightage to previous years. This helps determine how the companies have behaved during testing times and hence achieved a more meaningful risk score. The model can be improved by incorporating the latest financial data, which would include any performance improvements or declines that are going to be a key factor in predicting bankruptcy. In addition, the sample can be enlarged, which will increase the accuracy of the industry averages and provide results with higher confidence. Finally, the model can be applied to other industries, such as the pharmaceutical, technology, or energy sectors.

**Author Contributions:** Conceptualization, S.I.; methodology, S.I., S.K.J. and S.S.C.G.; software, S.K.J., S.I. and S.S.C.G.; validation, S.I., S.K.J. and G.B.; formal analysis, S.I., S.K.J. and G.B.; investigation, S.I., S.K.J.; resources, S.I., S.K.J. and S.S.C.G.; data curation, S.K.J. and S.S.C.G.; writing—original draft preparation, S.K.J., G.B. and S.S.C.G.; writing—review and editing, S.I.; visualization, S.I. and S.K.J.; supervision, S.I. and G.B.; project administration, G.B.; funding acquisition, G.B. All authors have read and agreed to the published version of the manuscript.

**Funding:** The Data Science Institute, Saint Peter's University.

**Data Availability Statement:** The data is available and can be emailed upon request.

**Conflicts of Interest:** The authors declare no conflict of interest.

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
