# Peer review of "A Comprehensive Approach to Bankruptcy Risk Evaluation in the Financial Industry"

_jrfm, doi:10.3390/jrfm17010041_

Round 1

Reviewer 1 Report

Comments and Suggestions for Authors

This paper proposes a comprehensive approach to evaluate the Bankruptcy Risk. My concern is how to determine the weights of different ratios. It is quite necessary the methodology. Another thing is that it is also necessary to discuss the reasons and implications behind the numbers in table 7. 

Author Response

Dear Referee:

We appreciate the many comments and suggestions you made to improve our paper. In the

new version of the paper we have tried to incorporate all your comments. In the attached report we detail some of the changes we have made in response to your comments.

Thanks again for your numerous suggestions.

Reviewer 2 Report

Comments and Suggestions for Authors

The authors have endeavoured to present a bankruptcy prediction metric using some ratios and developed risk profiles of only 20 financial sector companies in the USA. The authors claim it is useable in other sectors and has important policy implications for debt and profitability management.

While the authors put ample effort into building their model, it is not situated within a scholarly body of knowledge. There are plenty of bankruptcy prediction models in the finance literature, so the simple model presented in this model may not significantly impact the current practices. 

The paper has significant flaws that must be addressed before publishing it in this outlet:

(a) the introduction section is full of assertions, and the problem is not identified using the scholarly convention of 'problematization'. 

(b) the literature review section is presented but very little critique is used to persuade the readers to engage in this work. 

(c) the methodology section/findings are standalone pieces of works. In particular, the findings are not tied to the relevant scholarly works reviewed in the literature review section. This leaves the audience wondering why this paper should be read and where this work will lead them to think about corporate bankruptcy prediction! The authors must compare and contrast prior works and clearly demonstrate how their work has addressed the gaps in the prior work, outlined at the end of the literature review section. 

(d) the conclusion section needs to have reference to the research question/s. 

Overall, much effort is put into the methodology section, neglecting the introduction and the findings section. 

Author Response

(The authors gave the same response as above.)

Reviewer 3 Report

Comments and Suggestions for Authors

Dear Authors,

your manuscript is really interesting. 

Nevertheless, there is a strong need to improve your paper. 

I could not agree with assumption about weights:

"Weights considered for individual financial ratios within each category. "

I do not understand the methods You have used calculating weights 

"The following weights have been considered for the individual ratios within the fi- 187 nancial ratio categories in order to arrive at the categorical risk values as shown below in 188 Table 3" 

Altman's citation needs to be expanded to include more of his work. 

There is lack of limitations in Conclusions.

Please add Discussion. 

Author Response

(The authors gave the same response as above.)

Reviewer 4 Report

Comments and Suggestions for Authors

Give a more extensive explanation of the study's bankruptcy prediction model, including the precise factors and metrics utilized in the research.

To improve transparency and reproducibility of the research, provide more information about the data sources utilized in the study, including specific financial statements and time periods covered.

To offer a fuller view of the research, address any potential limitations of the study.

Author Response

(The authors gave the same response as above.)

Round 2

Reviewer 2 Report

Comments and Suggestions for Authors

The paper still has issues. I have commented on the changes made in the paper. Further comments:

1. Your model does not have a parent! It is your opinion only- not based on prior research. You need to have some references in support of the weighting scheme, as you claimed: "...This emphasis was informed by expert opinions that stress the critical role of liquidity in mitigating leverage risk, especially during economic downturns or financial crises. The combination of these approaches contributes to a well-rounded and contextually relevant weighting scheme tailored to the specific focus on leverage-related bankruptcy risk."

The model that you used to build your work is well discussed and elaborated. 

2. Ideally, you should outline some bankruptcy prediction methods/models in the second paragraph of your introduction and identify the problems in the model/models (with references). You will then address the problem/s you have identified and demonstrate how you are solving the problem. The use of Stein's (2012) model is a solution to your problem, but the audience is yet to learn your 'research problem or research question'. 

3. Looks okay.

4. Poor job done. You need references to all the discussions (at least the www sources or the newspaper clippings or some other form that you used to build the narratives). I could not find you tying the discussions to the literature you reviewed earlier in the literature review section. 

5. Well done. 

Other issues

Check your spelling and grammar in the next iteration. 

Author Response

Dear Referee:

We appreciate the additional comments and suggestions you made to improve our paper. In the

new version of the paper we have tried to incorporate all your comments. 

Reviewer 3 Report

Comments and Suggestions for Authors

Thank you for the amendments imposed. I have no further comments.

Author Response

Dear Referee:

We appreciate the comments and suggestions you made to improve our paper.